# Rapid Growth between 0 and 2 Years Old in Healthy Infants Born at Term and Its Relationship with Later Obesity: A Systematic Review and Meta-Analysis of Evidence

**DOI:** 10.3390/nu16172939

**Published:** 2024-09-02

**Authors:** Luz Doñate Carramiñana, Cristina Guillén Sebastián, Iris Iglesia Altaba, Carlos Nagore Gonzalez, Maria Luisa Alvarez Sauras, Sheila García Enguita, Gerardo Rodriguez Martinez

**Affiliations:** 1Department of Pediatrics, University of Zaragoza, 50012 Zaragoza, Spain; luzdonate@gmail.com (L.D.C.); gerard@unizar.es (G.R.M.); 2Instituto de Investigación Sanitaria Aragón (IISAragón), 50012 Zaragoza, Spain; cristinaguillen99@gmail.com (C.G.S.); mlalvarez@iisaragon.es (M.L.A.S.); gesheila@hotmail.com (S.G.E.); 3Primary Care Interventions to Prevent Maternal and Child Chronic Diseases of Perinatal and Developmental Origin Network (RICORS), RD21/0012/0012, Instituto de Salud Carlos III, 28029 Madrid, Spain; 4Growth, Exercise, Nutrition and Development(GENUD) Research Group, Instituto Agroalimentario de Aragón (IA2), University of Zaragoza, 50012 Zaragoza, Spain

**Keywords:** postnatal growth, rapid weight gain, obesity, overweight, programming

## Abstract

Introduction: Rapid growth in early childhood has been identified as a possible risk factor for long-term adiposity. However, there is a lack of studies quantifying this phenomenon only in healthy, full-term infants with appropriate birth weight for gestational age. This systematic review and meta-analysis aimed to investigate the association of rapid growth in full-term children up to 2 years of age with adiposity up to 18 years of age. Methodology: A systematic review of the literature was conducted in PubMed, EMBASE, and Web of Science. Results: 14 studies were included. We were unable to find strong evidence that rapid growth in early childhood is a risk factor for long-term adiposity. Rapid growth in early childhood was associated with taller heights (standardized mean difference: 0.51 (CI: 0.25–0.77)) and higher body mass index (standardized mean difference: 0.50 (CI: 0.25–0.76)) and a higher risk of overweight under 18 years. Conclusion: Rapid growth in early childhood in term infants with appropriate birth weight is associated with higher growth, body mass index, and risk of being overweight up to age 18, but further work is needed to identify the associations between early rapid growth and obesity later in adulthood.

## 1. Introduction

The global prevalence of obesity has increased exponentially in recent decades, with the current rate nearly tripling that of 1975 [1]. According to the World Obesity Federation, in 2020, there were 260 million children worldwide overweight and 175 million with obesity. Projections for 2035 estimate these numbers to rise to 390 million and 380 million, respectively, which would constitute 39% of the global child population. In 2020, Spain ranked as the twentieth country worldwide with the highest proportion of boys with overweight or obesity [2].

The International Obesity Task Force (IOTF) defines overweight as a body mass index (BMI) over 25 and obesity as a BMI over 30 kg/m² for children aged 2 to 19 years [3]. The cut-off points, according to the growth charts of the Centers for Disease Control and Prevention (CDC), are the 85th percentile for overweight and the 95th percentile for obesity [4]. Finally, the World Health Organization (WHO) uses the standard deviation (SD) of BMI to define childhood obesity in children aged 5 to 19 years. A BMI greater than one standard deviation above the WHO growth reference is considered overweight, and two standard deviations above is considered obesity, while for children under 5 years of age, the WHO considers >2SD as overweight and >3SD as obesity [5].

The role of social factors in the development of obesity has become increasingly significant in contemporary society. Variables such as socioeconomic status, cultural norms, and social networks profoundly influence dietary habits, physical activity levels, and access to healthcare, all of which are critical determinants of obesity. These social determinants can exacerbate disparities in obesity prevalence, particularly among marginalized groups.

Childhood obesity should be of concern mainly for two reasons. First, children who are overweight or obese are more likely to maintain this condition in adulthood compared to children with normal BMI. Second, childhood obesity leads to short- and long-term complications, including diseases that previously only affected adults, such as cardiovascular disease and type 2 diabetes mellitus [6]. By 2035, it is estimated that 68 million children will suffer metabolic disorders due to high BMI [2].

### Rapid Growth in Early Childhood

The etiology of non-monogenic obesity is multifactorial and is based on the complex interaction among genetic, environmental, and psychosocial factors. Many of these factors appear early in life (the first 1000 days) and are able to program the infant towards obesity [7,8]. Examples of this metabolic programming include the increased cardiovascular risk in adulthood among premature and low birth weight children, as well as the risk of developing overweight or obesity in childhood among those who experience rapid growth in the first two years of life [9,10].

The definition of rapid postnatal growth varies depending on the source consulted. A definition determines rapid growth as a change in the standard deviation of weight adjusted for age greater than or equal to +0.67 at any time between birth and 2 years [11]. This figure of 0.67 represents the width of the major percentile bands on a standard growth chart, so such an increase is interpreted as crossing at least one line on the chart [8,12].

The prevalence of rapid growth in the first two years of life ranges between 18% and 35% [13,14,15,16], although some studies estimate figures close to 50% [17,18]. Eighty-five percent of premature or low birth weight children experience rapid postnatal growth [19], while in full-term children with appropriate birth weight for gestational age, it occurs in 20–36% of cases [12,13,20,21,22].

Regarding breastfeeding, despite being considered the best source of nutrition for infants and children during the first year of life [23,24,25], there is controversy over whether it is a protective factor against obesity. The controversy surrounding whether breastfeeding prevents childhood obesity likely stems from the heterogeneity of the studies conducted. Variations in the definition of breastfeeding, as well as differences in the duration of breastfeeding and other cofactors such as lifestyle habits, complicate the interpretation of the results. This heterogeneity makes it challenging to draw definitive conclusions. Therefore, it is necessary to conduct studies with more homogeneous criteria in their definitions and study intervals to achieve clearer insights [26]. Formula feeding has been described as a risk factor for rapid postnatal growth [27,28,29,30]. Other risk factors for experiencing rapid postnatal growth include lower birth weight and gestational age, maternal tobacco consumption during pregnancy, and excessive weight gain during pregnancy.

Maternal overweightness during pregnancy has been associated with an increased risk of future overweight or obesity in offspring, potentially due to the heightened incidence of complications such as gestational diabetes. These complications can create an altered metabolic environment that may impact fetal development. Moreover, lifestyle differences, such as a lower prevalence of exclusive breastfeeding among mothers with overweight or obesity, may further contribute to the risk of childhood overweight [13]. Breastfeeding and the introduction of complementary feeding from 4 months onwards are considered protective factors [14,16,18,31].

Six previous systematic reviews evaluate the relationship between catch-up growth in early childhood and obesity or other measures of body composition in later stages, all concluding that there is a positive association [7,8,11,32,33,34]. However, none focus exclusively on full-term children with appropriate birth weight for gestational age. Therefore, this study conducted a systematic review to investigate the association of rapid growth in full-term infants up to 2 years of age on obesity and other measures of adiposity or body composition in later stages of life by summarizing findings from relevant studies and conducting a meta-analysis.

## 2. Materials and Methods

First, the International Prospective Register of Systematic Reviews (PROSPERO) was reviewed to check if there was an ongoing previous systematic review that fully addressed our topic, and the protocol for this review was registered (registration number CRD42024498743).

This systematic review has been designed following the guidelines of the PRISMA (Preferred Reporting Items for Systematic Reviews and Meta-Analyses) 2020 Statement [35]. 

Systematic searches were conducted in three electronic databases: PubMed, EMBASE, and Web of Science (WOS). The last search date was 22 April 2024. The search strategy was based on three groups of commands. Within each group, the “OR” operator was used, and between groups, the “AND” operator. The first group included terms related to the target population, namely, children and adolescents up to 18 years old; the second group consisted of terms related to adiposity; and the third group contained terms related to rapid growth and weight gain. Since the period of rapid growth to be analyzed spans from birth to 2 years of age, a fourth group of commands was used with children aged 0 to 2 years to ensure that this age period appeared in all articles retrieved in the search. Finally, filters for “Humans” and “Child: birth-18 years” were applied. In addition, terms from similar previous systematic reviews were reviewed to ensure that all relevant terms were included. The MESH terms and keywords used in the search are presented in Table 1.

### 2.1. Eligibility Criteria

The articles included in the review met the following inclusion criteria:(a)Studies conducted on children up to 18 years of age;(b)Studies that included data from patients born full-term (37–42 weeks of gestation) without previous pathology and with appropriate weight for gestational age;(c)Use of rapid postnatal growth as exposure, defined as a change in the standard deviation of weight adjusted for age greater than 0.67 at any time between birth and 2 years;(d)Analysis of at least one measure of adiposity (BMI, body fat percentage, trunk-to-body fat percentage ratio, fat mass, fat-free mass index, skinfold thickness, waist circumference, waist-to-height ratio, waist-to-hip ratio);(e)Results presented at ages between 2 and 18 years.

Exclusion criteria: No articles were excluded based on language, context, or time frame. Regarding the type of studies, only editorials, case descriptions, expert opinions, narrative reviews as well as incomplete papers were excluded.

### 2.2. Selection and Data Extraction

For the article selection process, two researchers (LDC and CGS) independently conducted selection at each phase (title, abstract, and full text) and discussed any discrepancies. A third researcher (IIA) intervened if consensus could not be reached regarding the inclusion of an article.

Once the articles to be included in this study were selected, the following data were extracted: last name of the main author, year and place of publication, study design and name, sample size, period of rapid growth studied, percentage of children in the sample who experienced it, adiposity variables investigated and age at which they were measured, statistical method used, identified covariables or confounding factors, and associations found between rapid postnatal growth and adiposity measures in later stages of life.

For observational studies, data from children who had experienced rapid postnatal growth were selected, and for experimental studies, data from the control group were chosen. Any missing items were sought in other associated articles or calculated from other data provided in the study.

### 2.3. Risk of Bias

As a tool to assess the risk of bias, the approach outlined in the JBI Manual for Evidence Synthesis for cohort studies was employed [36]. This tool evaluates the internal validity of studies by means of the following 11 items: 1. Were the two groups similar and recruited from the same population? 2. Were the exposures measured similarly to assign people to both exposed and unexposed groups? 3. Was the exposure measured in a valid and reliable way? 4. Were confounding factors identified? 5. Were strategies to deal with confounding factors stated? 6. Were the groups/participants free of the outcome at the start of the study (or at the moment of exposure)? 7. Were the outcomes measured in a valid and reliable way? 8. Was the follow-up time reported and sufficient to be long enough for outcomes to occur? 9. Was the follow-up complete, and if not, were the reasons for loss of follow-up described and explored? 10. Were strategies to address incomplete follow-up utilized? 11. Was appropriate statistical analysis used?

Each study was classified as high quality if it scored 7 or higher, acceptable quality if the score ranged from 4 to 6, or low quality if the score was 3 or lower. No articles were excluded based on their quality.

In order to identify potential publication bias in the meta-analyses, funnel plots were considered. However, due to the limited number of articles in each analyzed block and thus their diminished validity, it was ultimately decided not to include them in the final study.

### 2.4. Data Synthesis and Meta-Analysis

The data extracted from the different articles according to the previously indicated conditions, as well as their quality assessment, are presented in Table 2.

An evaluation of the extracted data was performed to conduct a meta-analysis. The articles used for this purpose were those providing mean and standard deviation data on height, weight, and BMI standard deviation, as well as those providing the incidence of overweight in the rapid postnatal growth group and the normal growth group. Meta-analyzing other variables and estimators was not possible due to the limited number of available articles.

The Jamovi software version 2.5.3 and version 2.3.28 module Major were used to carry out four meta-analyses, comparing children who experienced rapid growth in early childhood with those who did not.

First, a heterogeneity analysis was conducted using the restricted maximum likelihood method with tau2 and *I*^2^ estimators, as well as Cochrane’s Q test. Given the presence of heterogeneity, a random-effects model was employed [37]. The results of the meta-analysis are depicted using forest plots. The confidence interval of the average estimator was calculated using Cohen’s d. Statistical significance was set at *p* < 0.05 unless otherwise specified.

**Table 2 nutrients-16-02939-t002:** Characteristics and data synthesis of included studies (N = 14).

Author, Year, Country (City)	Study Characteristics	Rapid Growth Period % Rapid Growth	Age of Measurement and Adiposity Measurements	Statistical Method	Covariates	Results Rapid Growth (RG) vs. No Rapid Growth (NG) and/or Slow Growth (SG)	Quality
Ong et al., 2000, United Kingdom (Bristol) [38].	A prospective cohort study of 848 healthy full-term infants was randomly selected from 10% of the sample of another study (Avon longitudinal study of pregnancy and childhood) from July to December 1992.	0–24 months 30.7% (N = 260)	5 years Height SD, weight SD, BMI SD, % body fat (Brook and Siri equation), fat mass, waist circumference.	Mean ± SD or 95% CI of the difference between RG, NG, and SG. *p* < 0.05	Breastfeeding or artificial feeding	- SD height: RG 0.47 ± 0.80; NG 0.13 ± 0.89; SG −0.37 ± 0.98. - SD weight: RG 0.87 ± 0.93; NG 0.22 ± 0.87; SG −0.29 ± 0.93. - SD BMI: RG 0.82 ± 1.01; NG 0.19 ± 0.87; SG −0.07 ± 0.86. - % body fat: RG 17.2 (16.6–17.7); NG 15.8 (15.4–16.2); SG 14.7 (14.2–15.2). - Fat mass: RG 3.6 (3.4–3.7); NG 3.0 (2.9–3.1); SG 2.6 (2.5–2.8). - Waist circumference: RG 54.6 (54.2–55.1); NG 52.7 (52.3–53.0); SG 51.3 (50.9–51.8). *p* for all groups < 0.0005	High
Karaolis- Danckert et al., 2007, Germany (Dortmund) [12].	Open cohort study (Dortmund Nutritional and Anthropometric Longitudinally Designed Study) with 249 term healthy children with adequate gestational age size recruited since 1985.	0–24 months 28.5% (N = 71)	2–5 years Height SD, weight SD, BMI SD, % body fat (Deurenberg equation), tricipital fold SD, subscapular fold SD. Overweight (International Obesity Task Force, 10.04%, N = 25) Excess body fat (% body fat > 85th percentile, 18.47%, N = 46) Rate of change in % body fat and BMI between 2 and 5 years of age.	Mean ± SD or range (*p* < 0.05) of the difference between children with rapid growth and children without. β coefficients (*p* < 0.05).	-	- Height SD: RG 0.39 ± 0.98; NG −0.15 ± 0.96 (*p* < 0.0001). - Weight SD: RG 0.50 ± 0.94; NG −0.20 ± 0.95 (*p* < 0.0001). - BMI SD: RG 0.41 ± 0.90; NG −0.16 ± 0.99 (*p* < 0.0001). - % Body fat: RG 18 (15.4, 20.9); NG 16.7 (14.7, 19.4) (*p* = 0.02). - Triceps skinfold: SD RG 0.34 ± 0.94; NG −0.14 ± 0.99 (*p* = 0.0006). - Subscapular skinfold: SD RG 0.22 ± 1.08; NG −0.09 ± 0.96 (*p* = 0.03). - Overweight: RG 16.9% (12/71); NG 7.3% (13/178) (*p* = 0.02). - Excess body fat: RG 26.8% (19/71); NG 15.2% (27/178) (*p* = 0.03). - Trajectory of % body fat and BMI SD between 2 and 5 years: β 2.78, *p* < 0.0001; β 0.81, *p* < 0.0001.	High
Karaolis-Danckert et al., 2008, Germany (Berlin, Munich, Mainz, Düsseldorf, Freiburg) [13].	Longitudinal Cohort Study from Birth (German Multicenter Allergy Study) with 370 healthy full-term babies with gestational age recruited between January and December 1990 from 6 University Hospitals in 5 German cities (Berlin, Munich, Mainz, Düsseldorf, and Freiburg).	0–24 months 20% (N = 74)	2–6 years: Height SD, weight SD, BMI SD, % body fat (Slaughter equation), triceps skinfold SD, subscapular skinfold SD, overweight (International Obesity Task Force, 13.2%, n = 45/341). Rate of change in % body fat and BMI SD between ages 2 and 6 years.	Mean ± SD (*p* < 0.05) of the difference between children with rapid growth and children without. *p* of the difference in overweight rates between the two groups.		- Height SD: RG 0.74 ± 0.94; NG −0.13 ± 0.98 (*p* < 0.0001). - Weight SD: RG 0.85 ± 0.80; NG −0.14 ± 0.88 (*p* < 0.0001). - BMI SD: RG 0.62 ± 0.84; NG −0.10 ± 0.85 (*p* < 0.0001). - % body fat: RG 15.5 (13.0, 19.1); NG 13.2 (11.3, 16.2) (*p* = 0.0002). - Triceps skinfold: SD RG 0.48 ± 1.07; NG 0.12 ± 0.95 (*p* = 0.02). - Subscapular skinfold: SD RG 0.10 ± 1.20; NG −0.82 ± 1.20 (*p* < 0.0001). - Overweight: RG 28.2% (20/71); NG 9.3% (25/270) (*p* < 0.0001). - Trajectory % body fat and BMI SD between ages 2 and 6 years: β 1.83, *p* = 0.004; β 0.91, *p* = 0.0003.	High
Akaboshi et al., 2008, Japan (Kumamoto) [39].	Retrospective cohort study with 1353 healthy children born at term between 1988 and 2000 who were recruited between November 2003 and September 2004.	0–3/4 months 22.7% (N = 370)	3 years Height, Weight Overweight (International Obesity Task Strength, 4.73%).	Mean ± SD (*p* < 0.05) Odds ratios (95% CI, p) of being overweight	Sex, birth weight, BMI, breastfeeding, weight at 6–9 months and at 17–20 months, head and chest circumference at 3–4 months.	- Height SD RG 0.31 ± 0.91; NG 0.07 ± 0.88; SG −0.03 ± 0.84 (*p* < 0.0001) - Weight SD RG 0.53 ± 1.13; NG 0.14 ± 0.87; SG −0.03 ± 0.84 (*p* < 0.0001) OR boys (n = 697) 6.767 (2.180–21.007) *p* 0.0009; OR girls (n = 657) 4.966 (2.388–10.327) *p* < 0.0001 8.8% of children with RG were overweight, compared to 3.5% of children without.	High
Hui et al., 2008, China (Hong Kong) [40].	Prospective birth cohort study (Hong Kong Children of 1997 Birth Cohort) with 6075 healthy children born at term from April to May 1997, grouped into low, medium, and high birth weight.	0–3 months 3–12 months Growth rate tertiles 33.33% (N = 2025)	7 years SD BMI (WHO tables 2006), overweight and obesity (International Obesity Task Force, 15.3% n = 930).	Β coefficients (95% CI), Odds ratios (95% CI) of being overweight by sex and birth weight.	Sex, gestational age, baseline weight (at birth or 3 months), growth rate.	- BMI SD: 0–3 months β 0.50 (0.46–0.53), 3–12 months β 0.33 (0.28–0.37). - Overweight OR: RG 0–3 m girls (N = 899) 1.96 (1.22–3.15), boys (N = 882) 3.98 (2.62–6.05); RG 3–12 m girls (N= 899) 2.54 (1.43–4.53), boys (N = 882) 4.71 (2.86–7.77).	High
Larnkjaer et al., 2010, Denmark (Copenhagen) [21].	Cohort study prospective (Copenhagen Cohort Study on Infant Nutrition and Growth) with 95 healthy children born at term between 1987 and 1988 with adequate size for their gestational age	Change in standard deviation between 0–9 months and 0–3 months 23% (N = 22)	10 and 17 years Height, weight, BMI, % body fat, % trunk fat, waist-hip ratio, triceps skin fold	β coefficients (*p* < 0.05)	Sex, birth weight, BMI of parents	- 0–9 months. 17 years: BMI (β 1.223, *p* = 0.001), % body fat (β 1.388, *p* 0.028), % trunk fat (β 1.478, *p =* 0.025), triceps skinfold (β 1.824, *p* = 0.008). Waist–hip ratio (β 0.016, *p* = 0.082). No association with % trunk fat between % body fat (*p* = 0.414). - 0–3 months. 17 years: BMI (β 1.319, *p* = 0.029), % body fat (β 1.971, *p* 0.044), % trunk fat (β 2.096, *p* = 0.041), triceps skinfold (β 2.464, *p* = 0.023), waist-hip ratio (β 0.033, *p* = 0.018). No association between RG 3–6 m or 6–9 m.	High
Weng, 2013, United Kingdom [41].	Longitudinal cohort study (Millennium Cohort Study) with 13,513 healthy children born at term.	0–12 months Derivation Cohort: 42.9% (N = 3268) Validation Cohort: 42.9% (N = 772).	3 years Overweight (International Obesity Task Strength, 23.4%).	β coefficients (integer score) Odds ratios (95% CI) *p* < 0.05.	Sex, birth weight, paternal and maternal BMI before pregnancy, smoking during pregnancy (yes/no), breast at some point in the 1st year (yes/no).	OR 4.15 (3.64–4.73, *p* = 0.001) *p* = 0.05 β 1.4239 (integer score 19)	High
Ejlerskov et al., 2015, Denmark (Copenhagen) [42].	Prospective Cohort Study (SKOT) with 233 healthy full-term infants selected from the Danish national civil registry between April 2007 and May 2008.	0–5 months 15.9% (N = 37)	3 years Overweight (International Obesity Task Force 8.2%, WHO 1.7%), risk of being overweight (age-adjusted BMI SD > 1SD, 19.7%), age-adjusted BMI SD, fat mass index, fat-free mass index, skin folds.	β coefficients (*p* < 0.05) % of children with RG in the 4th quartile of fat mass index and free fat mass index (*p* < 0.05)	Parents: height and weight, weight gain during pregnancy, smoking during pregnancy, educational level. Child: gestational age, birth weight, feeding pattern (duration of breastfeeding, age of introduction of solids).	- Height: β 0.7 (n = 153, *p* < 0.05) - Weight: β 0.78 (n = 155, *p* < 0.001) - BMI: β 0.64 (n = 152, *p* < 0.001 ) - Fat mass index: β 0.42 (n = 152, *p* < 0.001) - Fat-free mass index: β 0.2 (n = 152, *p* < 0.001) - Fold sum: β 1.43 (n = 147) (*p* < 0.001) 6 months of exclusive breastfeeding eliminates this association. 48.7% of children with RG are in the 4th quartile of the fat mass index (N = 18), which is a % higher than that of children without (*p* < 0.031). The RG did not affect the probability of being in the highest quartile of fat-free mass index.	High
Wang, et al., 2016, United States [43].	Prospective cohort study from birth with 1442 children (926 full-term and 516 early-term)	0–4 months Slow (<−0.67), Normal (−0.67–0.67), Rapid (0.67–1.28), and Extremely Rapid (>1.28) 40.22% (N = 580)	2–7 Years Overweight or obesity (CDC; 40.22% N = 580) BMI SD.	Mean ± SD β coefficients (95% CI) Odds ratios for overweight or obesity (95% CI) *p* < 0.05	Maternal educational level, race, tobacco use during pregnancy, parity, pre-pregnancy BMI, hypertension, diabetes, fetal growth, and breastfeeding.	Full-Term (n = 926) RG and BMI Z-Score (0.97 ± 1.10, β = 0.37 (0.15, 0.59), *p* < 0.01). OR 1.5 (1.0, 2.4). RG and BMI Z-Score (1.04 ± 1.18, β = 0.39 (0.14, 0.64), *p* < 0.01). OR 1.6 (1.0, 2.6). Early-Term (N = 516) RG and BMI Z-Score (0.72 ± 1.27, β = 0.34 (0.06, 0.62), *p* < 0.05). OR 1.7 (1.0, 3.0). RG and BMI Z-Score (1.06 ± 1.15, β = 0.71 (0.45, 0.98), *p* < 0.001). OR 3.2 (1.9, 5.5) *p* < 0.001.	Acceptable
Flores-Barrantes et al., 2020, Spain (Zaragoza, Huesca, Teruel) [30].	Prospective Cohort Study from Birth (CALINA Study) with 767 healthy full-term children born between March 2009 and February 2010 in Aragón.	0–6 months 66.3% (N = 179)	3, 5, and 6 years Age-adjusted BMI Z-Score Age-adjusted weight Z-Score Age-adjusted height Z-Score	*p* < 0.05 Mediation analysis (type of feeding in the first 120 days): β coefficients (95% CI)	Child: Birth weight, gestational age Parents: educational level of both parents, paternal and maternal pre-pregnancy BMI, parents’ origin (Spanish or immigrant), tobacco use during pregnancy.	Trajectory of BMI Z-Score by age from birth to 6 years was greater in children with RG (*p* = 0.001). There was no association with the trajectory of height-for-age Z-Score (*p* = 0.89) or weight-for-age Z-Score (*p* = 0.16). At 6 years, BMI Z-Score for age (n = 767). β 0.784 (0.579–0.990). The type of feeding did not mediate this association.	High
Petrov et al., 2020, United States (Arizona, Texas) [22].	Secondary analysis of data from a clinical trial involving obese Mexican women recruited in the third trimester of pregnancy, with 126 healthy full-term infants with appropriate size for gestational age.	0-6 months 35.7% (N = 45)	36 months Overweight (CDC, 42.3%, N = 41/96)	Chi-square test (*p* < 0.05) Odds ratios (95% CI)	Mothers: Height and weight before pregnancy, parity, educational level, weight gain during pregnancy, tobacco use before pregnancy, assigned group in the trial. Child: Birth weight, sleep patterns.	N = 96 Chi-square test = 4.05, *p* = 0.04 OR = 2.35 (1.02–5.42)	High
Q. Lin et al., 2021, China (Shanghai) [44].	Birth Cohort Study (The Shanghai Sleep Birth Cohort Study) with 262 pregnant women were recruited at Renji Hospital from May 2012 to July 2023, along with their healthy full-term born children.	0–3 months 62% (N = 137)	4 years Weight, BMI, waist circumference, biceps circumference, subcutaneous fat.	β coefficients (95% CI)	Family income, gestational age, maternal BMI before pregnancy and paternal, RN weight in the first 3 days, energy intake at 6 months. 4 food scales, outdoor time, social media time, and sleep time.	Rapid Growth (RG): Weight β 0.90 (0.37–1.44), BMI β 0.93 (0.49–1.37), waist circumference β 1.90 (0.68–3.12), biceps circumference β 1.05 (0.62–1.48), subcutaneous fat β 2.57 (1.13–4.01). No relationship with waist-to-height ratio β 0.01 (0.00–0.02). Trajectory of Change in Weight-for-Age Z-Score: Weight β 0.83 (0.30–1.36), BMI β 0.82 (0.38–1.26), waist circumference β 1.70 (0.47–2.92), biceps circumference β 0.92 (0.50–1.35), subcutaneous fat β 2.25 (0.81–3.69). No relationship with waist-to-height ratio β 0.01 (0.00–0.02).	High
Fujita et al., 2021, Japan [45].	Retrospective cohort study (the Japan Kids Body-composition Study (JKB Study) with 423 adolescents born full-term.	0–18 months 33.17% (boys, n = 68) and 38.53% (girls, n = 84)	13–14 years BMI, waist circumference, fat mass index, % body fat	Mean of least squares standard error (*p* < 0.05)	Gestational age, sedentarism, rapid growth, maternal age in pregnancy, the appearance of pubic hair and puberal development, and birth weight adjusted by gestational age.	Boys: BMI RG 19.7 ± 0.4; NG 18.2 ± 0.4 (*p* < 0.01). Waist circumference RG 70.3 ± 1.2; NG 66.0 ± 1.1 (*p* < 0.01). Fat mass index RG 3.1 ± 0.2; NG 2.4 ± 0.2 (*p* < 0.01). Body Fat Percentage RG 15.2 ± 0.8; NG 12.9 ± 0.7 (*p* < 0.01). Girls: BMI RG 20.8 ± 0.4; NG 19.8 ± 0.4 (*p* = 0.01). Waist circumference RG 70.5 ± 1.0; NG 68.3 ± 0.9 (*p* = 0.02). Fat mass index RG 5.3 ± 0.2; NG 4.7 ± 0.2 (*p =* 0.01). Body fat percentage RG 24.5 ± 0.7; NG 22.9 ± 0.7 (*p* = 0.03).	High
Taylor et al. 2023, New Zealand (Dunedin) [46].	Secondary analysis of data from a randomized controlled trial in the first years of life with 341 full-term babies >2500 g	0–24 months (gaps of 6 or 12 months) 0.7–24.3%	11 years Obesity (WHO; 18.5%).	Positive RG predictive value for obesity at age 11.	Sex, ethnicity, level of deprivation, maternal education, assigned group.	Percentage of children with RG who were obese at 11 years: 0–6 months: 31.4%, 6–12 months: 22%, 12–18 months: 23.5%, 18–24 months: 33.3%, 0–12 months: 23.9%, 6–18 months: 17.6%, 12–24 months: 27.6%.	High

Note: SD: standard deviation. 95% CI: 95% confidence interval. BMI: body mass index. WHO: World Health Organization. RG: Rapid growth, NG: No rapid growth, SG: Slow growth.

## 3. Results

### 3.1. Selection of Studies

During the initial search phase, 6296 articles were retrieved (5226 from PubMed, 693 from EMBASE, and 377 from Web of Science). After removing duplicate studies, 5767 articles remained, which underwent screening based on title and abstract. Following the title review, 5374 irrelevant articles were excluded, and after abstract screening, 246 articles were excluded from the review. Finally, 141 articles underwent full-text screening, with 129 of them being excluded. Additionally, the bibliographic references of the seven systematic reviews retrieved in the search were reviewed, and two new studies were selected, resulting in a total sample of 14 studies presented in Table 2. The flow diagram illustrating the study selection process is shown in Figure 1.

### 3.2. Characteristics of the Studies

From the 14 included studies, 12 were cohort studies and 2 were secondary data analyses from a randomized clinical trial [22,46]. Within the cohort studies, 10 were prospective studies [12,13,21,30,38,40,41,42,43,44], and 2 were retrospective [39,45]. The studies were published between 2000 and 2023 and were conducted in 8 countries across 4 continents: United Kingdom [38,41], Germany [12,13], Denmark [21,42], Japan [39,45], China [40,44], United States [22,43], Spain [30], and New Zealand [46]. They include a total sample of 26,097 participants, with a range of sample sizes between 95 [21] and 13,513 [41]. 

Participants in all studies were healthy, full-term-born children. In 5 of the 14 studies, full-term born children (≥ 39 weeks gestation) were included [22,43,44,45,46], and one of them performed a stratified analysis according to gestational age (full-term and early term) [43]. The period of measured rapid postnatal growth varies among studies, although it always had a limit of two years of age.

### 3.3. Risk of Bias

Using the checklist for cohort studies from the “JBI Manual for Evidence Synthesis”, 13 studies were classified as high quality and 1 study as acceptable quality [43]. No study was rated as low quality. The authors declared no conflicts of interest in all articles except for one, in which this information was not provided [39]. Several of the studies reported limited generalizability of their results due to a small or non-representative sample [12,13,21,22,30,42,44,45,46].

### 3.4. Results of Individual Studies

All studies described a positive direct relationship between rapid postnatal growth and adiposity variables in the later stages of childhood. More detailed information on the analyzed studies, their variables, and the results obtained in each can be found in Table 2.

### 3.5. Synthesis of Meta-Analysis Results and Sensitivity Analysis

#### 3.5.1. Standardized Mean Differences of the Standard Deviation of Height

Four valid studies were obtained for the mean and standard deviation of height from the six articles that addressed it, all classified as high quality. The random-effects model was used, given the results in the heterogeneity analysis (Q = 18.27; *p* = 0.0004, tau² = 0.0591, *I*² = 85.99%). Figure 2 shows the forest plot of the meta-analysis.

The estimated average standardized mean difference (SMD) was 0.51 (CI: 0.25–0.77; *p* = 0.0001). The results remained very similar after excluding the study with the highest sample weight by Akaboshi et al. [39], with the average SMD in this case being 0.60 (CI: 0.31–0.89) (Appendix A).

#### 3.5.2. Standardized Mean Differences of the Standard Deviation of Weight

Four valid studies were obtained from the six articles dealing with mean and standard deviation of the standard deviation of weight, all considered high quality. The random-effects model was used, given the results in the heterogeneity analysis (Q = 26.37; *p* < 0.0001, tau² = 0.0787, *I*² = 88.85%). Figure 3 shows the forest plot of the meta-analysis. The estimated average SMD was 0.74 (CI: 0.44–1.03; *p* < 0.0001). After excluding the study with the highest sample weight (Akaboshi et al. [39] with a weight of 27.24%), the results maintained the same direction. The estimated average mean difference was 0.86 (CI: 0.59–1.12) (Appendix A).

#### 3.5.3. Standardized Mean Differences of the Standard Deviation of BMI

Five valid studies were extracted from the six articles dealing with this topic. The random-effects model was used, given the results in the heterogeneity analysis (Q = 26.16; *p* < 0.0001, tau² = 0.0708, *I*² = 84.63%). Figure 4 shows the forest plot of the meta-analysis.

In the study by Wang et al., differences were not statistically significant for either of the two groups analyzed [43], while in the studies carried out by Ong et al. [38] and Karaolis-Danckert et al. [12,13], the differences found were statistically significant. The estimated average SMD was 0.50 (CI: 0.25–0.76) with a *p*-value of 0.0001. After excluding the study with the highest weight in the analysis (Ong et al. [38] with a weight of 21.84%), the average mean difference was 0.45 (CI: 0.14–0.76) (Appendix A).

#### 3.5.4. Logarithmic Odds Ratios for Developing Overweight

Data were extracted from five articles classified as high quality, except for Wang et al. [43], which was rated acceptable. The random-effects model was used, given the results in the heterogeneity analysis (Q = 11.16; *p* = 0.0248, tau² = 0.0913, *I*² = 68.67%). Figure 5 shows the forest plot of the meta-analysis.

The estimated average logarithmic odds ratio was 0.75 (CI: 0.41–1.09) with *p* < 0.0001, and the odds ratio was 2.12 (95% CI: 1.51–2.98). The exclusion of the study with the highest weight in the analysis (Wang et al. [43], with a sample weight of 28.04%) did not change the direction of the results. The estimated average logarithmic odds ratio became 0.88 (95% CI 0.54–1.22) (Appendix A), with the odds ratio being 2.41 (95% CI 1.71–3.39). The heterogeneity analysis showed altered results, indicating that it was not statistically significant (Q = 4.95; *p* = 0.1752, tau² = 0.0516, *I*² = 42.98%).

## 4. Discussion

The results of this systematic review and meta-analysis support the evidence from previous reviews regarding the association between rapid growth in early childhood and the risk of obesity, as well as with BMI, weight, height, body composition measures, waist circumference, and skinfold thickness in later stages of life, even when excluding the preterm group [7,8,11,32,33,34].

The value of this review compared to existing literature lies in its focus on children born at term with appropriate weight for gestational age, incorporating various body composition parameters, not just the risk of overweight/obesity. Additionally, in the reviews conducted by Zheng et al. [8], Baird et al. [32], Andrea et al. [33], and Ong et al. [7], the relationship between postnatal catch-up growth and obesity was not limited to childhood but extended into adulthood. Moreover, Baird et al. [32], Andrea et al. [33], Ong et al. [7], Halilagic et al. [11], and Chen et al. [34] did not restrict included articles based on the definition of rapid growth (change in weight-for-age SD score > 0.67 within any time during the first two years of life). Andrea et al. [33] focused on studies involving ethnic or low socioeconomic populations.

### 4.1. Height, Weight, and Body Mass Index

Height, weight, and body mass index were also influenced by rapid growth in the first months of life. Children who experienced rapid growth in the first two years of life were, on average, 0.51 standard deviations taller (CI: 0.25–0.77) and 0.74 standard deviations heavier (CI: 0.44–1.03) than those who did not.

Additionally, their BMI was 0.5 standard deviations higher (CI: 0.25–0.76), indicating an increase in long-term adiposity. The meta-analysis by Chen et al. in 2020 also demonstrated the association between rapid postnatal growth and BMI in later stages (weighted mean difference 0.57 [CI: 0.36–0.79]), although it included preterm children and those with low or inadequate birth weight for gestational age [34]. Similarly, a combined analysis of seven cohort studies in 2023 established a relationship between rapid growth in early childhood and BMI standard deviation [47].

### 4.2. Risk of Overweight/Obesity

Children who experienced rapid growth during any period of the first two years of life were twice as likely to be overweight in later stages compared to children who did not experience rapid growth (estimated OR 2.122 [CI: 1.510–2.982]). This figure was lower than that reported by the systematic review and meta-analysis by Zheng et al. in 2008 (OR 3.66 [CI: 2.59–5.17]) [8]; also lower than the combined analysis of seven cohort studies in 2023 (OR 4.49 [CI: 3.61–5.59]) [47]; and higher than that estimated by a non-systematic meta-analysis of 10 cohort studies in 2012 (OR 1.97 [CI: 1.83–2.12]), where the cut-off point for defining rapid growth was 1 standard deviation instead of 0.67 [48]. None of these three studies exclude children born prematurely or with low or inadequate birth weight.

Although low birth weight and prematurity have been associated with rapid postnatal growth and obesity due to increased insulin resistance and body fat deposition in this group [8], our findings in term newborns with appropriate weight for their gestational age confirm that the relationship between rapid growth and obesity is more complex than previously described. These results highlight the need for future studies to elucidate the pathophysiological mechanisms underlying this relationship.

### 4.3. Adiposity Measurements

Although no quantitative synthesis of other outcomes has been conducted due to the variety of adiposity measures studied and the limited number of studies examining each of them, positive associations were found between rapid postnatal growth and increase in adiposity measures later in life, such as body fat percentage [12,13,21,38,45], fat mass [38,42,45], skinfold thickness [12,13,21,42], and waist circumference [38,44,45]. These findings support conclusions reached in two systematic reviews published in 2018 and 2020 [8,11]. Similarly, a meta-analysis published in 2020 suggested that catch-up weight in early childhood had a positive correlation with body fat percentage later in life compared to children without catch-up [34].

There are three variables for which no association with rapid postnatal growth was demonstrated. These include the trunk-to-body fat percentage ratio [21], the waist-to-height ratio [44], and the likelihood of being in the highest quartile of fat-free mass index [42]. Only a slight association was found with the waist-to-hip ratio [21]. However, each of these variables has been analyzed in only one study, limiting the evidence available for drawing solid conclusions.

The mechanisms of rapid growth have so far been clarified, and some authors defend that adipose tissue plays an important role in signaling rapid growth, favoring fat deposition through biomarkers such as leptin and associating low levels of it with rapid growth by inhibiting satiety [34]. This mechanism would partially justify the relationship between rapid growth and the relationship found with increased adiposity; however, it does not seem to be sufficient by itself, so a deeper understanding of rapid growth is necessary to avoid future complications and diseases associated with increased adiposity.

### 4.4. Period and Duration of Postnatal Rapid Growth

The periods of rapid growth studied in the works included in the review vary widely, but all fall within the first two years of life. Regarding overweight and obesity, it was observed that they were related to postnatal rapid growth regardless of the chosen period, but the strength of this association varied. In this respect, the first three months of life seemed to be the ones that had the greatest relationship with the programming effect in the future.

One study examined two consecutive time periods (0–3 months and 3–12 months) and found that the risk of obesity increased the later rapid growth occurs (OR 0–3 months for girls: 1.96 and for boys: 3.98; OR 3–12 months for girls: 2.54 and for boys: 4.71); however, the association with BMI weakened as the studied interval progressed (β 0–3 months: 0.50; β 3–12 months: 0.33) [40]. Another study investigated the relationship of rapid growth with BMI, body fat percentage, trunk fat percentage, and triceps skinfold thickness in three consecutive time periods (0–3 months, 3–6 months, 6–9 months), finding a positive result only when rapid growth occurred in the first three months of life [21]. Zheng et al.’s 2018 meta-analysis concluded that rapid growth during the first year of life has a greater impact on adiposity measures than growth up to two years (OR: 4.12 vs. OR: 3.58) [8], and Halilagic et al.’s 2020 systematic review determined that the first three months are of greatest importance within the first year [11]. In contrast, a meta-analysis published in 2020 concluded that postnatal rapid growth was not related to BMI and body fat percentage in later stages when it lasted less than two years [34].

### 4.5. Covariates

These associations could be mediated by numerous maternal and child factors, with speculation that one of them could be birth weight. Infants with low birth weight or are small for gestational age are more likely to experience postnatal rapid growth than infants with normal weight, along with increased insulin resistance and central fat deposition, making them more vulnerable to weight gain [49,50,51]. However, in this review, all included articles involved children with birth weights greater than 2500 g and appropriate for gestational age, or analyses were adjusted for birth weight as a covariate. Thus, it could be stated that children with appropriate birth weight who experienced postnatal rapid growth also suffer long-term increases in adiposity.

Nutrition in early childhood plays a significant role in obesity risk and body composition in children. A cohort study with term-born children of appropriate birth weight demonstrated that breastfeeding acts as a protective factor in the effect of postnatal rapid growth on childhood obesity risk [12]. Another recent study revealed that formula milk is a risk factor for both experiencing rapid growth in early childhood and having a higher BMI at age 6, although it does not mediate the association between postnatal rapid growth and childhood obesity [30]. A 2015 study found that breastfeeding for 1–3 months and 4–5 months slightly attenuates the effect of rapid growth from birth to five months on body fat index in later stages compared to breastfeeding for less than one month, while breastfeeding for 6 months eliminates the effect [42]. Exclusive breastfeeding reduces the growth rate and likely the risk of developing overweight in children and adolescents compared to formula feeding. Formula milk is associated with greater growth, possibly due to its higher protein content or because its consumption is linked to higher energy intake [29,52]. In our meta-analysis, we did not include it as a covariate because not all analyzed studies reported this variable.

Additionally, some maternal factors such as pre-pregnancy nutritional status, smoking, educational level, and socioeconomic status are related to both postnatal rapid growth and obesity [14,16], but similar to what happens with feeding behaviors, due to the heterogeneity in reported studies, these variables were not included in the analyses.

Based on these findings and expert recommendations, although the diagnosis of rapid growth remains challenging, it is crucial to carefully evaluate the outcomes, assess the underlying causes of such growth in each individual, and consider the potential benefits of weight gain on a case-by-case basis, as in the instance of catch-up growth following significant faltering growth [53].

## 5. Strengths

This review was conducted following a rigorous and systematic methodology under the guidelines of the PRISMA statement [35], being the first to evaluate the relationship between rapid postnatal growth and adiposity in later stages of life in healthy children born at term with appropriate weight for gestational age.

A total of 71.43% (10/14) of the included studies were prospective cohort studies, providing the highest level of evidence within the group of observational studies. Additionally, all included studies, except one, have been rated as high quality and low risk of bias, further strengthening the results obtained. The variety of countries in which the included studies were conducted suggests that the effect of rapid postnatal growth is not limited to certain population groups. The literature search was not restricted by language, reducing the likelihood that relevant studies were excluded from the review.

## 6. Limitations

The main limitation is the heterogeneity among the included studies. The diversity of methodologies, definitions of rapid growth, and study variables, along with the variability in defining pediatric obesity and the application of its criteria, add further heterogeneity to the analysis, making direct comparison and synthesis of results difficult. To alleviate this problem, future studies should strive for greater consistency in the definition and measurement of rapid growth and obesity.

Another limitation is the lack of inclusion of some key variables, such as feeding patterns, socioeconomic status, and other environmental factors that can have a significant impact on child development and obesity risk. In addition, some analyses, such as visceral fat indices or fat-free mass, are based on a very limited number of studies, making it difficult to draw firm conclusions. Future studies should include larger samples to better understand the impact of rapid growth on different aspects of health in later life.

## 7. Conclusions

This systematic review and subsequent meta-analysis confirm that rapid postnatal growth also occurs in children born at term with appropriate weight for gestational age and is associated with the risk of overweight, as well as with weight, height, and body mass index in later stages of life. Additionally, it is related to measures of adiposity and body composition, such as body fat percentage, fat mass, waist circumference, and skinfold thickness. This update of the available evidence emphasizes the need for anthropometric measurements in pediatric follow-ups to detect and address rapid postnatal growth, as well as the importance of developing protocols or guidelines based on the study of its underlying mechanisms and the prevention of its risk factors, with the aim of preventing future childhood obesity and its associated morbidity.

## Figures and Tables

**Figure 1 nutrients-16-02939-f001:**
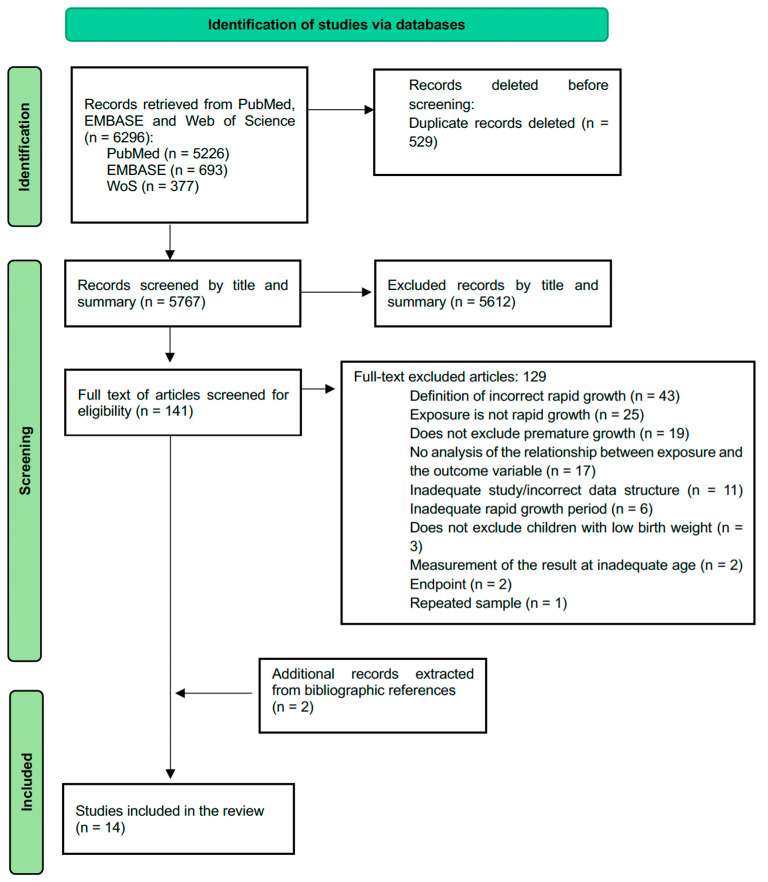
Flow diagram according to the PRISMA declaration.

**Figure 2 nutrients-16-02939-f002:**
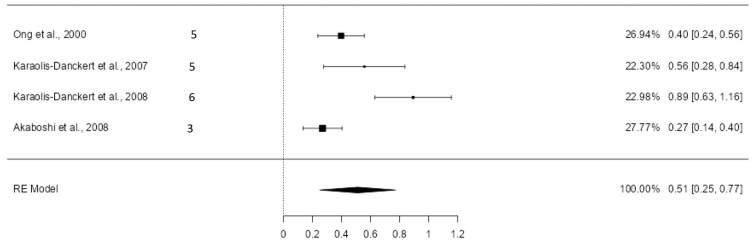
Forest plot of the analysis of the mean difference in standard deviation of height [12,13,38,39].

**Figure 3 nutrients-16-02939-f003:**
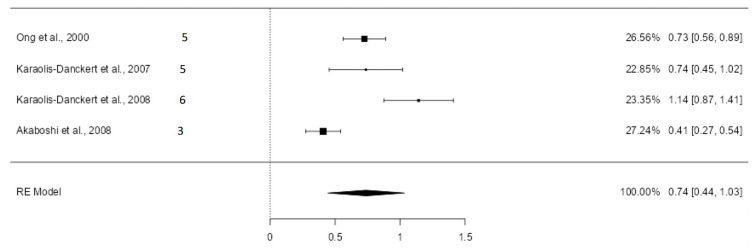
Forest plot of the analysis of the mean difference in standard deviation of weight [12,13,38,39].

**Figure 4 nutrients-16-02939-f004:**
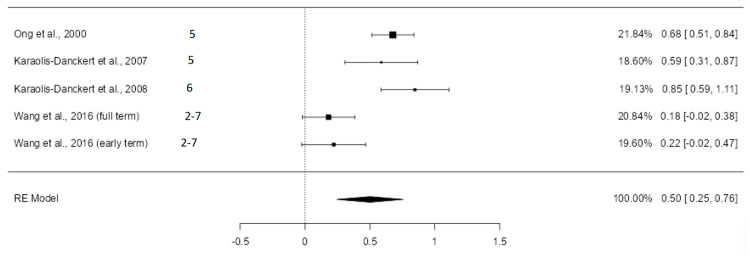
Forest plot of the analysis of the mean difference in standard deviation of BMI [12,13,38,43].

**Figure 5 nutrients-16-02939-f005:**
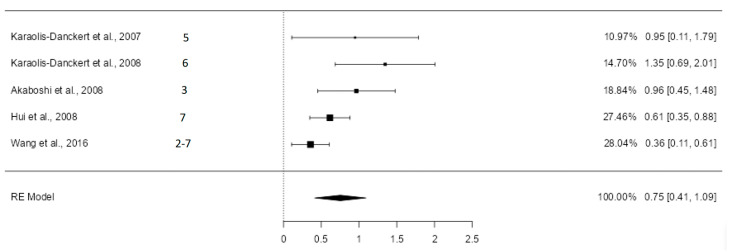
Forest plot of the analysis of the difference in logarithmic odds ratios [12,13,39,40,43].

**Table 1 nutrients-16-02939-t001:** MESH terms and keywords used in the systematic review.

Number	Term	Limit
1.1	“Infant, newborn” OR “Infant” OR “Child, preschool” OR “Child” OR “Adolescent”	MeSH Terms
1.2	“Infants” OR “Newborn*” OR “Neonate *” OR “Preschool *” OR “Child*” OR “Infancy”	Title/Abstract
2.1	“Obesity” OR “Pediatric Obesity” OR “Obesity, abdominal” OR “Overweight” OR “Adiposity” OR “Adipose Tissue” OR “Body Fat Distribution” OR “Waist Circumference” OR “Body Mass Index” OR “Skinfold Thickness”	MeSH Terms
2.2	“Obes *” OR “Overweight” OR “Adipos *” OR “Fat” OR “Fatty” OR “Waist Circumference” OR “Body Mass Index” OR “Quetelet index” OR “BMI” OR “Skinfold thickness *”	Title/Abstract
3	“Weight” OR “Catch-up” OR “Rapid Weight Gain” OR “RWG” OR “Rapid Growth”	Title/Abstract
4.1	“Infant, newborn” OR “Infant”	MeSH Terms
4.2	“Infants” OR “Newborn *” OR “Neonate *” OR “Infancy”	Title/Abstract
5	1.1 OR 1.2	
6	2.1 OR 2.2	
7	4.1 OR 4.2	
8	3 AND 5 AND 6 AND 7	Filters: Humans, Child: birth-18 years

## Data Availability

All the information used is available to the public. The data extracted from the studies and any other material used in the review are presented in the different sections of the work, in the annexes, or referenced in the bibliography.

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
