# Peer review of "Rapid Growth between 0 and 2 Years Old in Healthy Infants Born at Term and Its Relationship with Later Obesity: A Systematic Review and Meta-Analysis of Evidence"

_nutrients, 2024, doi:10.3390/nu16172939_

Round 1

Reviewer 1 Report

Comments and Suggestions for Authors

This manuscript reviewed and analyzed the relationship between rapid growth during the first two years of life in healthy infants born at term and later obesity. The topic is of interest, contextualization is appropriated, the text well-structured and the reading easy to follow.  Thus, I suggest some minor changes to improve the current manuscript:

-       Line 96: “The last search date was March 2024”. Maybe it would be interesting indicating also the day.

-       Line 115: “Analysis of at least one measure of adiposity”. Please, indicate which were the measures of adiposity.

-       Table 1: Weng , 2013. Please, consider including the number of individuals in the third column (although this study is longitudinal) to homogenize.

-       In discussion section, main results are separately addressed and provide clear information. However, I miss some sentences where authors hypothesize about possible causes, physiological mechanisms, consequences or implications of these results; which may be useful for further research in this area.

Author Response

Thank you very much for your corrections and suggestions. We have already applied the necessary changes to the manuscript. We indicate where each correction has been implemented in the manuscript:

  1. Line 96: “The last search date was March 2024”. Maybe it would be interesting indicating also the day.

Line 116: We have added “The last search date was 22 April 2024”.

  1. Line 115: “Analysis of at least one measure of adiposity”. Please, indicate which were the measures of adiposity.

Line 136: We have added the main measures of adiposity utilized: BMI, body fat percentage, trunk-to-body fat percentage ratio, fat mass, fat-free mass index, skinfold thickness, waist circumference, waist-to-height ratio, waist-to-hip ratio.

  1. Table 1: Weng , 2013. Please, consider including the number of individuals in the third column (although this study is longitudinal) to homogenize.

Table 1, Weng, 2013: We have added the number of individuals “Derivation Cohort: 42.9% (N= 3268), Validation Cohort: 42.9% (N= 772)”.

  1. In discussion section, main results are separately addressed and provide clear information. However, I miss some sentences where authors hypothesize about possible causes, physiological mechanisms, consequences or implications of these results; which may be useful for further research in this area.

We have added two paragraphs indicating about possible causes, physiological mechanisms, and consequences of our results:

Line 334 “Although low birth weight and prematurity have been associated with rapid postnatal growth and obesity due to increased insulin resistance and body fat deposition in this group [8], our findings in term newborns with appropriate weight for their gestational age confirm that the relationship between rapid growth and obesity is more complex than previously described. These results highlight the need for future studies to elucidate the pathophysiological mechanisms underlying this relationship”.

Line 355 “The mechanisms of rapid growth have so far been clarified and some authors defend that adipose tissue plays an important role in signaling rapid growth, favoring fat deposition through biomarkers such as leptin, associating low levels of it with rapid growth by inhibiting satiety [34]. This mechanism would partially justify the relationship between rapid growth and the relationship found with increased adiposity; however, it does not seem to be sufficient by itself, so a deeper understanding of rapid growth is necessary to avoid future complications and diseases. associated with increased adiposity”.

Your feedback has greatly enhanced the quality of our work, and we appreciate your thorough review.

Reviewer 2 Report

Comments and Suggestions for Authors

Nutrients: Rapid growth between 0 and 2 years old in healthy infants born at term and its relationship with later obesity 3125901

Review:

While their question of the association between rapid growth in full-term children up to 2 years of age with adiposity 18 later in life can be a valid question, the underlying question about whether rapid growth between 0 and 2 years affects later obesity is ideally addressed looking at weight status in adulthood. The abstract conclusion suggests that the outcomes are inevitable, which is not appropriate, rather the word “associated” should be used. The abstract conclusion should end with: Further work is needed to identify the associations between early rapid growth with obesity in adulthood.

Reference 11 says that “The common definition of rapid growth during infancy is a change in weight or length-for-age standard deviation score greater than +0.67 from birth to age 24 months”, it does not say is it the “most widely used” common so these words should be omitted and replaced with “a”.  A better definition to base this study is >1 SD as recommended by this expert guideline: Cooke 36976274 Catch-up growth in infants and young children with faltering growth - Expert opinion to guide general clinicians 2023

Since body composition measures were not meta-analyzed, body composition should not be mentioned in the abstract.

Weight for age is not a useful measure as taller people should weight more than those with shorter height, so that “weigh more” in the abstract conclusion should be omitted.

The limited strength of the associations, with confidence intervals should be reported in the abstracts as well as measures of heterogeneity.

Since “identified covariables or confounding factors” these should be mentioned in the text and abstract in a discussion of possible causal models.

I don’t understand what looks like confidence intervals expanded on the forest plots’ graphic summary, as they do not show agreement with the confidence intervals beside them. The height & weight for age were statistically and clinically significantly higher. I do not understand whether BMI is statistically and clinically significantly different because the confidence interval includes the null value. The forest plots should include the age of the outcome measures in a column since if all of these studies reported the outcome at age 3, the meaning of these results is less important.

Observational studies are seldom high quality so I encourage the authors to use a risk of bias assessment that includes an examination of confounding and not to judge the studies as high quality too readily.

The sentence “All studies describe a direct relationship between rapid postnatal growth and adiposity variables in later stages of childhood” – could mean two things: a positive relationship (as opposed to a negative one) or a strong relationship. This needs clarifying.

Differences in growth as well as differences in overweight can be due to the determinants of health, which is an important point that needs to be made in the text and abstract. The introduction should describe the social determinants of overweight to less blame the victims.

This paper should be cited: Cooke 36976274 Catch-up growth in infants and young children with faltering growth - Expert opinion to guide general clinicians since the tone of this paper is judgemental towards heavier weights and it is important that it be recognized that accelerated growth can be difficult to accurately diagnose for individuals and children should not be encouraged to experience faltering growth because of misplaced fears about potentially inducing accelerated growth.

Why are log odds ratios reported? I think the academic and clinical community understands odds ratios better than log odds (although risk ratios would be a stronger metric if possible to use). The statistics shown on the figures are standardized mean differences in standard deviations, which could and likely should be reported as abbreviated to SMD SDS for standardized mean differences in standard deviation scores. A statistician should be consulted.

In a sensitivity analysis, the largest study was excluded. There is little rationale for doing so. It would make sense to assess risk of bias and exclude the most biased study instead and/or to limit the analysis to outcomes at ages 15 years and above. As I see that most of the studies were of young children, this analysis should be stratified in the forest plots and the odds ratios to <5 years, 6-7 years and 11 to 14 years, 15-17 years. There is much less concern if the strength of the association diminishes with age, but currently that aspect cannot be analyzed.

It is not appropriate to assume homogeneity when the p-value for heterogeneity is non-significant or whatever reason was used. It would be better to quote the Cochrane Collaboration for wording for the assessment of the heterogeneity results.

Good systematic review practice includes listing the references of the 129 excluded studies in a supplemental file.

The text including the aim should be written in the past tense and since this is looking at observational studies, the word “impact” should be replaced with “association”.

The fonts in the figures are too small to read them easily.

The introduction lists several categorizations of overweight and obesity, which are sometimes used inappropriately. For example, some use the WHO categorizations for >5 years for children under age 5, where better knowledge likely leads to less misclassification. This paper should comment if any of the references make this error and analyze the result using the WHO’s guidance instead of repeating the error if any is found (Ref: de Onis Defining obesity risk status in the general childhood population: which cut-offs should we use? 2010 PMID: 20233144)

Author Response

Thank you very much for your corrections and suggestions. We have already applied the necessary changes to the manuscript. We indicate where each correction has been implemented in the manuscript:

  • While their question of the association between rapid growth in full-term children up to 2 years of age with adiposity 18 later in life can be a valid question, the underlying question about whether rapid growth between 0 and 2 years affects later obesity is ideally addressed looking at weight status in adulthood. The abstract conclusion suggests that the outcomes are inevitable, which is not appropriate, rather the word “associated” should be used. The abstract conclusion should end with: Further work is needed to identify the associations between early rapid growth with obesity in adulthood.

Line 25: We have added: Rapid postnatal growth in the first two years of life in term children with birth-appropriate weight for gestational age is associated with higher height, higher weight and BMI and have a higher risk of overweight before age 18, which could have consequences for their future health. Further work is needed to identify the associations between early rapid growth with obesity in adulthood.

  • Reference 11 says that “The common definition of rapid growth during infancy is a change in weight or length-for-age standard deviation score greater than +0.67 from birth to age 24 months”, it does not say is it the “most widely used” common so these words should be omitted and replaced with “a”.A better definition to base this study is >1 SD as recommended by this expert guideline: Cooke 36976274 Catch-up growth in infants and young children with faltering growth - Expert opinion to guide general clinicians 2023.

Line 69: “A definition determines rapid growth as a change in standard deviation of weight adjusted for age greater than or equal to +0.67 at any time between birth and 2 years”.

While it is true that catch-up growth is typically defined as an increase of 1 SD, in this study, we focused on rapid growth in newborns with appropriate weight for gestational age who did not experience significant weight loss. Therefore, we believe it is more appropriate to refer to this phenomenon as accelerated growth rather than catch-up growth. Consequently, we consider the definition of +0.67 SD to be more suitable for this context.

  • Since body composition measures were not meta-analyzed, body composition should not be mentioned in the abstract.

Line 20: We have deleted the sentence “based on the body composition variables.”

  • Weight for age is not a useful measure as taller people should weight more than those with shorter height, so that “weigh more” in the abstract conclusion should be omitted.

We have utilized standardized weight deviations, already adjusted for both sex and age, as they are one of the most commonly used parameters in pediatric populations to assess nutritional status, as is done in multiple studies for this purpose.

  • The limited strength of the associations, with confidence intervals should be reported in the abstracts as well as measures of heterogeneity. 

We agree that this information is of interest and relevant to the abstract. However, due to the word limit restrictions, we were unable to include it. If the editors consider it appropriate to increase the word count to accommodate this information, we would be willing to apply such a change.

  • Since “identified covariables or confounding factors” these should be mentioned in the text and abstract in a discussion of possible causal models.

The covariates are already mentioned and discussed in detail in section 4.5 of the Discussion. While it would indeed be interesting to explore this topic further, the length of the article is already extensive. We believe that a more in-depth analysis would be better suited for future publications.

  • I don’t understand what looks like confidence intervals expanded on the forest plots’ graphic summary, as they do not show agreement with the confidence intervals beside them. The height & weight for age were statistically and clinically significantly higher. I do not understand whether BMI is statistically and clinically significantly different because the confidence interval includes the null value. The forest plots should include the age of the outcome measures in a column since if all of these studies reported the outcome at age 3, the meaning of these results is less important.

The confidence intervals referred to by the reviewer probably refer to the meta-analyses conducted after the exclusion of certain studies. However, the intervals presented in the manuscript correspond to the confidence intervals prior to the exclusion of those studies with the largest samples. We consider it more appropriate to present them this way, as the information following the exclusion is complementary to the initial analysis.

The confidence interval of the BMI does not include the value null (0).

Age has been included in the column of the figures.

  • Observational studies are seldom high quality so I encourage the authors to use a risk of bias assessment that includes an examination of confounding and not to judge the studies as high quality too readily. 

Line 160: We have added the criteria to explain the quality score applied to studies analysed from the JBI Manual for Evidence Synthesis for cohort studies.

  • The sentence “All studies describe a direct relationship between rapid postnatal growth and adiposity variables in later stages of childhood” – could mean two things: a positive relationship (as opposed to a negative one) or a strong relationship. This needs clarifying.

Line 230: We have added “All studies describe a positive direct relationship”

  • Differences in growth as well as differences in overweight can be due to the determinants of health, which is an important point that needs to be made in the text and abstract. The introduction should describe the social determinants of overweight to less blame the victims.

Line 48: We have added: “The role of social factors in the development of obesity has become increasingly significant in contemporary society. Variables such as socioeconomic status, cultural norms, and social networks profoundly influence dietary habits, physical activity levels, and access to healthcare, all of which are critical determinants of obesity. These social determinants can exacerbate disparities in obesity prevalence, particularly among marginalized groups”.

  • This paper should be cited: Cooke 36976274 Catch-up growth in infants and young children with faltering growth - Expert opinion to guide general clinicians since the tone of this paper is judgemental towards heavier weights and it is important that it be recognized that accelerated growth can be difficult to accurately diagnose for individuals and children should not be encouraged to experience faltering growth because of misplaced fears about potentially inducing accelerated growth.

Line 411: We have added “Based on these findings and expert recommendations, although the diagnosis of rapid growth remains challenging, it is crucial to carefully evaluate the outcomes, assess the underlying causes of such growth in each individual, and consider the potential benefits of weight gain on a case-by-case basis, as in the instance of catch-up growth following significan faltering growth [53]” as well as the paper in the References: “53.          Cooke R, Goulet O, Huysentruyt K, Joosten K, Khadilkar AV, Mao M, Meyer R, Prentice AM, Singhal A. Catch-Up Growth in Infants and Young Children With Faltering Growth: Expert Opinion to Guide General Clinicians. J Pediatr Gastroenterol Nutr. 2023 Jul 1;77(1):7-15”.

  • Why are log odds ratios reported? I think the academic and clinical community understands odds ratios better than log odds (although risk ratios would be a stronger metric if possible to use). The statistics shown on the figures are standardized mean differences in standard deviations, which could and likely should be reported as abbreviated to SMD SDS for standardized mean differences in standard deviation scores. A statistician should be consulted. 

We appreciate your observation regarding the comprehensibility of odds ratios (OR) in result interpretation. However, we have chosen to use log-odds ratios (log-OR) in our meta-analysis for several statistical reasons. Log-ORs provide a more symmetric distribution, which better aligns with the normality assumptions underlying many statistical methods, and they simplify the process of combining results across studies by converting multiplications and divisions into additions and subtractions. Additionally, the variance of log-ORs tends to be more stable, enhancing the precision of the analysis.

  • In a sensitivity analysis, the largest study was excluded. There is little rationale for doing so. It would make sense to assess risk of bias and exclude the most biased study instead and/or to limit the analysis to outcomes at ages 15 years and above. As I see that most of the studies were of young children, this analysis should be stratified in the forest plots and the odds ratios to <5 years, 6-7 years and 11 to 14 years, 15-17 years. There is much less concern if the strength of the association diminishes with age, but currently that aspect cannot be analyzed.

We excluded the study with the highest weight in our meta-analysis as part of a sensitivity analysis to assess the robustness and reliability of our results. This approach helps to determine whether our overall findings are disproportionately influenced by a single study, and it allows us to evaluate potential biases that could skew the meta-analytic conclusions. By removing the most influential study, we ensure that our results are not unduly dependent on it and that the observed effects are consistent across the remaining studies.

  • It is not appropriate to assume homogeneity when the p-value for heterogeneity is non-significant or whatever reason was used. It would be better to quote the Cochrane Collaboration for wording for the assessment of the heterogeneity results. 

In assessing heterogeneity, we have utilized the random-effects model. This model assumes that the studies included in the meta-analysis are not identical but rather represent a distribution of effects across different populations or settings. It accounts for both within-study and between-study variability, making it particularly useful for evaluating heterogeneity. By considering the variability among study results, the random-effects model provides a more generalized estimate of the effect size, reflecting the true variation across the studies. We consider that this approach is appropriate for our analysis as it allows for a more accurate assessment of the heterogeneity present in the included studies.

  • Good systematic review practice includes listing the references of the 129 excluded studies in a supplemental file. 

We have added the full list of 129 excluded studies as an annex.

  • The text including the aim should be written in the past tense and since this is looking at observational studies, the word “impact” should be replaced with “association”.

Line 17: We have added “This review and meta-analysis aimed to investigate the association of rapid growth in full-term children up to 2 years of age with adiposity later in life”.

  • The fonts in the figures are too small to read them easily.

We agree that these aspects are crucial for readers understanding. However, we would like to note that the final design and formatting of the tables are in hands of the editorial team. We have no objections to implement them as required.

  • The introduction lists several categorizations of overweight and obesity, which are sometimes used inappropriately. For example, some use the WHO categorizations for >5 years for children under age 5, where better knowledge likely leads to less misclassification. This paper should comment if any of the references make this error and analyze the result using the WHO’s guidance instead of repeating the error if any is found (Ref: de Onis Defining obesity risk status in the general childhood population: which cut-offs should we use? 2010 PMID: 20233144).

We agree with the reviewer's suggestion and have addressed this concern in the revised manuscript. Specifically, we have added the variability in the definition and application of obesity criteria as a limitation of our study:

Line 430: “The diversity of methodologies, definitions of rapid growth, and study variables, along with the variability in defining pediatric obesity and the application of its criteria, add further heterogeneity to the analysis, making direct comparison and synthesis of results difficult. To alleviate this problem, future studies should strive for greater consistency in the definition and measurement of rapid growth and obesity”.

Your feedback has greatly enhanced the quality of our work, and we appreciate your thorough review.

Reviewer 3 Report

Comments and Suggestions for Authors

This is a generally well written paper on an important topic which in my experience is not well known or understood amongst many health care professionals, nutritionists or dietitians.

There  are number of changes/suggestions/comments that I would ask the authors to consider.

Page 2 Line 72. The authors state that there is debate to whether or nor breast feeding protects against later obesity.  This is true, but I think the reader would benefit from a few more sentences on this topic, maybe citing a few of the “for or against” published articles/studies.

Page 7. Table 2.  Personally I found this table difficult to read easily, especially the key outcomes found in the extreme right column described a s “Results”.  I would be grateful if the authors could consider if there is another way of creating the table.?

Figures 2,3,4 and 5.  I think presenting the percentage data to 2 decimal places is excessive. Please consider reporting to just one decimal place.

Figure 4.  I am not sure why the term “termino complets” appears on two occasions.

Overall this manuscript represents a significant amount of work by the authors. I feel the manuscript will be of use to many working in the field.

Author Response

Thank you very much for your corrections and suggestions. We have already applied the necessary changes to the manuscript. We indicate where each correction has been implemented in the manuscript:

1- Page 2 Line 72. The authors state that there is debate to whether or nor breast feeding protects against later obesity.  This is true, but I think the reader would benefit from a few more sentences on this topic, maybe citing a few of the “for or against” published articles/studies.

We have added a paragraph explaining the difficulty in assessing the role of breastfeeding in preventing obesity due to the heterogeneity of studies.

Line 81: “The controversy surrounding whether breastfeeding prevents childhood obesity likely stems from the heterogeneity of the studies conducted. Variations in the definition of breastfeeding, as well as differences in the duration of breastfeeding and other cofactors such as lifestyle habits, complicate the interpretation of the results. This heterogeneity makes it challenging to draw definitive conclusions. Therefore, it is necessary to conduct studies with more homogeneous criteria in their definitions and study intervals to achieve clearer insights [26]”.

2- Page 7. Table 2.  Personally I found this table difficult to read easily, especially the key outcomes found in the extreme right column described a s “Results”.  I would be grateful if the authors could consider if there is another way of creating the table.?

Thank you for your suggestions regarding the clarity and interpretation of the tables. We agree that these aspects are crucial for readers understanding. However, we would like to note that the final design and formatting of the tables are in hands of the editorial team. We have no objections to implement them as required.

3- Figures 2,3,4 and 5.  I think presenting the percentage data to 2 decimal places is excessive. Please consider reporting to just one decimal place.

 The research team uses the Jamovi statistical program, which does not allow for the option of displaying results with only one decimal place.

4- Figure 4.  I am not sure why the term “termino complets” appears on two occasions.

We have deleted the term “termino complets” from figure 4.

Your feedback has greatly enhanced the quality of our work, and we appreciate your thorough review.

Reviewer 4 Report

Comments and Suggestions for Authors

The article ‘Rapid growth between 0 and 2 years old in healthy infants born at term and its relationship with later obesity: a systematic review and meta-analysis of evidence’ presents a valuable review of studies on rapid growth in children in the first two years of life and its relationship with later obesity. This work was conducted in accordance with PRISMA guidelines, demonstrating a high methodological standard. Most of the included studies are prospective cohort studies, which further enhances the evidentiary value of the results. The inclusion of studies from different countries increases the universality of the results and indicates the global nature of the problem.

Nevertheless, this article has several important limitations that need to be taken into account before publication. The most important limitation is the considerable heterogeneity between studies, which may affect the interpretation of the results. The diversity of methodologies, definitions of rapid growth and study variables makes direct comparison and synthesis of results difficult. To alleviate this problem, future studies should strive for greater consistency in the definition and measurement of rapid growth.

Another limitation is the lack of inclusion of some key variables, such as feeding patterns, socioeconomic status and other environmental factors that can have a significant impact on child development and obesity risk. Future analyses would benefit from including these variables to provide a more complete picture of the factors influencing outcomes. In addition, some analyses, such as visceral fat indices or fat-free mass, are based on a very limited number of studies, making it difficult to draw firm conclusions. Future studies should include larger samples to better understand the impact of rapid growth on different aspects of health in later life.

In conclusion, the article provides substantial evidence on the association between rapid growth in the first two years of life and later risk of obesity and overweight. However, the authors should consider suggested improvements, such as increasing methodological consistency, including more confounding variables, to fully understand these relationships and develop effective childhood obesity prevention strategies. Only after these changes have been made can the manuscript be accepted for publication.

Author Response

We appreciate your feedback regarding the study's limitations. We agree with the points you raised and have incorporated them into the "Limitations" section of the manuscript. Specifically, these changes have been added at line 430: “The diversity of methodologies, definitions of rapid growth, and study variables, along with the variability in defining pediatric obesity and the application of its criteria, add further heterogeneity to the analysis, making direct comparison and synthesis of results difficult. To alleviate this problem, future studies should strive for greater consistency in the definition and measurement of rapid growth and obesity.

Another limitation is the lack of inclusion of some key variables, such as feeding patterns, socioeconomic status and other environmental factors that can have a significant impact on child development and obesity risk. In addition, some analyses, such as visceral fat indices or fat-free mass, are based on a very limited number of studies, making it difficult to draw firm conclusions. Future studies should include larger samples to better understand the impact of rapid growth on different aspects of health in later life”.

Your comments have helped to strengthen the overall rigor of our work, and we are grateful for your contribution.

Reviewer 5 Report

Comments and Suggestions for Authors

Dear Authors,

Thank you for your manuscript. The paper is well-written and presents important knowledge summarizing evidence on child overweight and obesity in relation to rapid growth during the first two years of life. Please see my comments below.

In the Introduction section, the risk factors of infant overweight and obesity are well presented. However, I would suggest adding evidence of maternal gestational diabetes and high birth weight in relation to overweight and obesity in later child life.

Lines 60-65. What about the change of height supporting the definition of rapid child growth? Because further results include studies presenting the change in height as well.

Line 115. I think measures of adiposity should be listed and summarized here (fat mass, fat percentage, BMI, WC, W to H ratio, skinfold thickness, etc.).

Lines 117-119. What about excluding incomplete papers like conference proceedings and protocol papers containing no results?

Section 2.2. Was any software assistance used for paper extraction and screening (e.g., Covidence)?

Section 2.3. The criteria for scoring paper quality should be explained.

The results are clearly summarized and well-discussed.

Comments on the Quality of English Language

Minor English editing is required if accepted.

Author Response

Thank you very much for your corrections and suggestions. We have already applied the necessary changes to the manuscript. We indicate where each correction has been implemented in the manuscript

1- In the Introduction section, the risk factors of infant overweight and obesity are well presented. However, I would suggest adding evidence of maternal gestational diabetes and high birth weight in relation to overweight and obesity in later child life.

We have added a paragraph to explain the influence of obesity and gestational diabetes on the development of childhood obesity:

Line 91: “Maternal overweight during pregnancy has been associated with an increased risk of future overweight or obesity in offspring, potentially due to the heightened incidence of complications such as gestational diabetes. These complications can create an altered metabolic environment that may impact fetal development. Moreover, lifestyle differences, such as a lower prevalence of exclusive breastfeeding among mothers with overweight or obesity, may further contribute to the risk of childhood overweight [13]”.

2- Lines 60-65. What about the change of height supporting the definition of rapid child growth? Because further results include studies presenting the change in height as well.

While we acknowledge that there is some variability in the definitions of rapid growth, we have opted to use the definition based on weight, as it is the most commonly employed in the literature. This choice allows for greater consistency with previous studies and facilitates comparison of our findings with existing research.

3- Line 115. I think measures of adiposity should be listed and summarized here (fat mass, fat percentage, BMI, WC, W to H ratio, skinfold thickness, etc.).

Line 136: We have added the main measures of adiposity utilized: BMI, body fat percentage, trunk-to-body fat percentage ratio, fat mass, fat-free mass index, skinfold thickness, waist circumference, waist-to-height ratio, waist-to-hip ratio.

4- Lines 117-119. What about excluding incomplete papers like conference proceedings and protocol papers containing no results?

Line 142: We have added: “as well as incomplete papers were excluded”.

5- Section 2.2. Was any software assistance used for paper extraction and screening (e.g., Covidence)?

No paper extraction or screening assistance software has been used.

6- Section 2.3. The criteria for scoring paper quality should be explained.

We have added the criteria to explain the quality score applied to studies analysed from the JBI Manual for Evidence Synthesis for cohort studies:

Line 160: “This tool evaluates the internal validity of studies by means of the following eleven items: 1. Were the two groups similar and recruited from the same population?; 2. Were the exposures measured similarly to assign people to both exposed and unexposed groups?; 3. Was the exposure measured in a valid and reliable way?; 4. Were confounding factors identified?; 5. Were strategies to deal with confounding factors stated?; 6. Were the groups/participants free of the outcome at the start of the study (or at the moment of exposure)?; 7. Were the outcomes measured in a valid and reliable way?; 8. Was the follow up time reported and sufficient to be long enough for outcomes to occur?; 9. Was follow up complete, and if not, were the reasons to loss to follow up described and explored?; 10. ¿ Were strategies to address incomplete follow up utilized?; 11. Was appropriate statistical analysis used?”

 Your comments have helped to strengthen the overall rigor of our work, and we are grateful for your contribution.

Round 2

Reviewer 2 Report

Comments and Suggestions for Authors

“Finally, the World Health Organization (WHO) uses the standard deviation (SD) of BMI to define childhood obesity in children aged 5 to 19 years. A BMI greater than one standard deviation above the WHO growth reference is considered overweight, and two standard deviations above is considered obesity [5].” The second sentence about the WHO is not correct as written as it needs to include “5 to 19 years” to be accurate and ideally the authors need to point include that for under 5 years of age the WHO considers >2 SD as overweight and >3 SD obesity.

I’m concerned that the included forest plots are a subset and not the complete ones. The authors responded: “The confidence intervals referred to by the reviewer probably refer to the meta-analyses conducted after the exclusion of certain studies. However, the intervals presented in the manuscript correspond to the confidence intervals prior to the exclusion of those studies with the largest samples”. The figures included in a paper should be more complete and show the most important information, before and after the exclusion of the largest studies. Now that journals will publish an unlimited number of figures in supplementary documents, there is no reason to exclude figures or include only non-representative figures. 

In the results about BMI:
“In the study by Wang et al., differences were not statistically significant for either of the two groups analyzed [42]. The estimated average SMD is 0.50 (CI: 0.25 - 0.76) with a p-value of 0.0001. After excluding the study with the highest weight in the analysis (Ong  et al. [39] with a weight of 21.84%), the average mean difference is 0.45 (CI: 0.14 - 0.76).” Why is one and only one study featured in the sentence before the meta-analysis results?

It is important for these authors compare their BMI meta-analysis results with the WHO recommendations for the appropriate age, in an assessment of clinical importance so not to rely only on statistical significance. They report in their abstract that “‘Results: 14 studies were included from the initial 5767 retrieved. The meta-analyses demonstrated a positive association between rapid postnatal growth and height, weight, body mass index (BMI), and overweight.’, however it should be put into context in both the paper and abstract: a SMD difference of 0.50 would not be classified as overweight for either WHO age group. Combined with the greater height of their exposed group, these results suggest that the more rapidly growing infants are more likely to be taller people with slightly higher BMIs but not to a pathological degree.

A valid abstract (that is not overly judgemental about overweight) is:

Introduction: Rapid growth in early childhood has been identified as a possiblerisk factor for long- term adiposity. However, there is a lack of studies quantifying this phenomenon in healthy full-term children with appropriate birth weight for gestational age. This systematic review and meta-analysis aimed to investigate the association of rapid growth in full-term children up to 2 years of age with adiposity up to 18 years of age. Methodology: A systematic review of the literature was conducted in PubMed, EMBASE, and Web of Science. Results: 14 studies were included. We were unable to find evidence that rapid growth in early childhood is a risk factor for long-term adiposity. Rapid growth in early childhood was associated with taller heights (standardized mean difference (SMD): 0.51 (CI: 0.25 - 0.77) and higher BMIs (SMD: 0.50 (CI: 0.25 - 0.76)) and a higher risk of overweight under 18 years. Conclusion: Further work is needed to identify the associations between early rapid growth with obesity in adulthood.

The paper now appropriately mentions the determinants of health and refers to risks of bias but I do not see that these authors examined and reported how well each study considered the possible confounding by the determinants of health. They have included some information in the Tables but have not given this important topic thought or synthesis. Study quality is mentioned briefly in the log odds section but not in reference to the BMI meta-analysis.

It is not appropriate to assume homogeneity “indicating homogeneity” when the p-value for heterogeneity is non-significant or based on the I2 value or something else. This is invalid, similar to the error of stating that two groups are similar when they are found not statistically different.

“Similarly, a meta-analysis published in 2020 asserts that children with rapid growth in early childhood have a higher body fat percentage later in life compared to children without rapid growth [34]” Did they find evidence to support this point or just “assert” this point? This is an important concern. Also did those meta-analysis authors take confounding by possible common causes of both rapid growth in early childhood and higherbody fat percentage later in life?

Line 343 “strong positive associations have been found” either needs the references cited that are being referred to or it needs to be revised.

The paper should be written in the past tense throughout.

Comments on the Quality of English Language

none

Author Response

  • “Finally, the World Health Organization (WHO) uses the standard deviation (SD) of BMI to define childhood obesity in children aged 5 to 19 years. A BMI greater than one standard deviation above the WHO growth reference is considered overweight, and two standard deviations above is considered obesity [5].” The second sentence about the WHO is not correct as written as it needs to include “5 to 19 years” to be accurate and ideally the authors need to point include that for under 5 years of age the WHO considers >2 SD as overweight and >3 SD obesity.

Line 47: We have added “while for children under 5 years of age the WHO considers >2 SD as overweight and >3 SD obesity”.

  • I’m concerned that the included forest plots are a subset and not the complete ones. The authors responded: “The confidence intervals referred to by the reviewer probably refer to the meta-analyses conducted after the exclusion of certain studies. However, the intervals presented in the manuscript correspond to the confidence intervals prior to the exclusion of those studies with the largest samples”. The figures included in a paper should be more complete and show the most important information, before and after the exclusion of the largest studies. Now that journals will publish an unlimited number of figures in supplementary documents, there is no reason to exclude figures or include only non-representative figures.

We have added a complementary document with the forest plots indicated in the paper after the exclusion of certain studies.

  • In the results about BMI:
    “In the study by Wang et al., differences were not statistically significant for either of the two groups analyzed [42]. The estimated average SMD is 0.50 (CI: 0.25 - 0.76) with a p-value of 0.0001. After excluding the study with the highest weight in the analysis (Ong  et al. [39] with a weight of 21.84%), the average mean difference is 0.45 (CI: 0.14 - 0.76).” Why is one and only one study featured in the sentence before the meta-analysis results?

We mentioned only the studies by Wang et al. because they were the only ones where no statistically significant differences were observed. However, to address your concern and improve clarity, we have added the following phrase to line 275: "while in the studies carried out by Ong et al. (39) and Karaolis-Danckert et al. (12,13), the differences found were statistically significant." This addition clarifies the contrast between the results of these studies and those by Wang et al.

  • It is important for these authors compare their BMI meta-analysis results with the WHO recommendations for the appropriate age, in an assessment of clinical importance so not to rely only on statistical significance. They report in their abstract that “‘Results: 14 studies were included from the initial 5767 retrieved. The meta-analyses demonstrated a positive association between rapid postnatal growth and height, weight, body mass index (BMI), and overweight.’, however it should be put into context in both the paper and abstract: a SMD difference of 0.50 would not be classified as overweight for either WHO age group. Combined with the greater height of their exposed group, these results suggest that the more rapidly growing infants are more likely to be taller people with slightly higher BMIs but not to a pathological degree.

It is correct that some studies did not use the WHO recommendations for categorizing and analyzing the sample in groups based on the proposed definition of obesity and overweight. We have already acknowledged this limitation in our study in line 433: "The diversity of methodologies, definitions of rapid growth, and study variables, along with the variability in defining pediatric obesity and the application of its criteria, add further heterogeneity to the analysis, making direct comparison and synthesis of results difficult. To alleviate this problem, future studies should strive for greater consistency in the definition and measurement of rapid growth and obesity."

Given the nature of a meta-analysis, we can draw conclusions from a group of studies, but we are not able to reanalyze results based on classifications not performed in those studies. The statement, "Results: 14 studies were included from the initial 5767 retrieved. The meta-analyses demonstrated a positive association between rapid postnatal growth and height, weight, body mass index (BMI), and overweight," is consistent with the meta-analysis conducted. We believe that healthcare professionals and researchers who read this study will be able to appropriately interpret the results and contextualize the conclusions within the framework of the study’s limitations.

  • A valid abstract (that is not overly judgemental about overweight) is:

Introduction: Rapid growth in early childhood has been identified as a possible risk factor for long- term adiposity. However, there is a lack of studies quantifying this phenomenon in healthy full-term children with appropriate birth weight for gestational age. This systematic review and meta-analysis aimed to investigate the association of rapid growth in full-term children up to 2 years of age with adiposity up to 18 years of age. Methodology: A systematic review of the literature was conducted in PubMed, EMBASE, and Web of Science. Results: 14 studies were included. We were unable to find evidence that rapid growth in early childhood is a risk factor for long-term adiposity. Rapid growth in early childhood was associated with taller heights (standardized mean difference (SMD): 0.51 (CI: 0.25 - 0.77) and higher BMIs (SMD: 0.50 (CI: 0.25 - 0.76)) and a higher risk of overweight under 18 years. Conclusion: Further work is needed to identify the associations between early rapid growth with obesity in adulthood.

We have applied the following changes to the abstract in order to adapt it to the observations made by the reviewer:

Abstract: Introduction: Rapid growth in early childhood has been identified as a possible risk factor for long-term adiposity. However, there is a lack of studies quantifying this phenomenon only in healthy, full-term infants with appropriate birth weight for gestational age. This system-atic review and meta-analysis aimed to investigate the association of rapid growth in full-term children up to 2 years of age with adiposity up to 18 years of age. Methodology: A systematic re-view of the literature was conducted in PubMed, EMBASE, and Web of Science. The results were analyzed in blocks based on the body composition variables. Meta-analyses were performed us-ing data on standard deviations (SD) of height, weight, body mass index, and the incidence of overweight up to 18 years. Results: 14 studies were included. We were unable to find strong evi-dence that rapid growth in early childhood is a risk factor for long-term adiposity. Rapid growth in early childhood was associated with taller heights (standardized mean difference: 0.51 (CI: 0.25 - 0.77) and higher body mass index (standardized mean difference: 0.50 (CI: 0.25 - 0.76)) and a higher risk of overweight under 18 years. Conclusion: Rapid growth in early childhood in term infants with appropriate birth weight is associated with higher growth, body mass index and risk of overweight up to age 18 but further work is needed to identify the associations between early rapid growth and obesity later in adulthood.

  • The paper now appropriately mentions the determinants of health and refers to risks of bias but I do not see that these authors examined and reported how well each study considered the possible confounding by the determinants of health. They have included some information in the Tables but have not given this important topic thought or synthesis. Study quality is mentioned briefly in the log odds section but not in reference to the BMI meta-analysis.

The complexity and scope of the review we conducted make it challenging to address every specific detail requested by the reviewer. Including such details would likely complicate the understanding of the study’s objective and the interpretation of the results. We would like to reiterate that covariates do indeed vary across the studies, adding to the heterogeneity. This effect has been acknowledged and reflected in the discussion and limitations sections of our study.

  • It is not appropriate to assume homogeneity “indicating homogeneity” when the p-value for heterogeneity is non-significant or based on the I2 value or something else. This is invalid, similar to the error of stating that two groups are similar when they are found not statistically different.

Line 293: We have added “indicating that it was not statistically significant”.

  • “Similarly, a meta-analysis published in 2020 asserts that children with rapid growth in early childhood have a higher body fat percentage later in life compared to children without rapid growth [34]” Did they find evidence to support this point or just “assert” this point? This is an important concern. Also did those meta-analysis authors take confounding by possible common causes of both rapid growth in early childhood and higherbody fat percentage later in life?

We agree that the conclusions of the studies should be interpreted with caution. Accordingly, we have made the suggested changes to the manuscript. Specifically, we have added the following statement in line 349: “Similarly, a meta-analysis published in 2020 suggests that catch-up weight in early childhood has a positive correlation with body fat percentage later in life compared to children without catch-up [34].” We believe this addition enhances the discussion and provides a more nuanced understanding of the results.

  • Line 343 “strong positive associations have been found” either needs the references cited that are being referred to or it needs to be revised.

Line 343: We have deleted “strong”. We have also added the references of the studies where these associations have been described: “positive associations were found between rapid postnatal growth and increases in adi-posity measures later in life, such as body fat percentage [12, 13, 21, 39, 46], fat mass [39, 41, 46], skinfold thickness [12, 13, 21, 41], and waist circumference [39, 43, 46]”

  • The paper should be written in the past tense throughout

The paper has been revised and changes made to be written in past tense.